# Suitability of Biodegradable Materials in Comparison with Conventional Packaging Materials for the Storage of Fresh Pork Products over Extended Shelf-Life Periods

**DOI:** 10.3390/foods9121802

**Published:** 2020-12-04

**Authors:** Luzia M. Hawthorne, Anel Beganović, Matthias Schwarz, Aeneas W. Noordanus, Markus Prem, Lothar Zapf, Stefan Scheibel, Gerhard Margreiter, Christian W. Huck, Katrin Bach

**Affiliations:** 1Department of Food Technology and Nutrition, Management Center Innsbruck, Universitaetsstrasse 15, 6020 Innsbruck, Tyrol, Austria; luziamh@outlook.com; 2Institute for Analytical Chemistry and Radiochemistry, Leopold Franzens University of Innsbruck, Innrain 80-82, 6020 Innsbruck, Tyrol, Austria; anel.beganovic@student.uibk.ac.at (A.B.); christian.w.huck@uibk.ac.at (C.W.H.); 3MULTIVAC Sepp Haggenmüller SE & Co. KG, Bahnhofstrasse 4, 87787 Wolfertschwenden, Bayern, Germany; matthias.schwarz@multivac.de (M.S.); stefan.scheibel@multivac.de (S.S.); 4NATURABIOMAT GmbH, Oberer Feldweg 64, 6130 Schwaz, Tyrol, Austria; noordanus@naturabiomat.at (A.W.N.); g.margreiter@naturabiomat.at (G.M.); 5Department for Food and Packaging Technology, Kempten University of Applied Sciences, Bahnhofstrasse 61, 87435 Kempten, Bayern, Germany; markus.prem@hs-kempten.de; 6ZLV-Zentrum für Lebensmittel- und Verpackungstechnologie e.V., Ignaz-Kiechle-Strasse 20-22, 87437 Kempten, Bayern, Germany; lothar.zapf@zlv.de

**Keywords:** modified atmosphere packaging, biodegradable packaging, meat quality, green packaging, meat shelf life

## Abstract

The packaging of fresh meat has been studied for decades, leading to improved packaging types and conditions such as modified atmosphere packaging (MAP). While commonly used meat packaging uses fossil fuel-based materials, the use of biodegradable packaging materials for this application has not been studied widely. This study aimed at evaluating the sustainability of biodegradable packaging materials compared to established conventional packaging materials through analyses of the quality of freshly packaged pork. The quality was assessed by evaluating sensory aspects, meat color and microbiological attributes of the pork products. The results show no significant differences (*p* > 0.05) in ground pork and pork loin stored in biodegradable MAP (BioMAP) and conventional MAP for the evaluated sensory attributes, meat color or total bacterial count (TBC) over extended storage times. The data suggest that BioMAP could be a viable alternative to MAP using conventional, fossil fuel-based materials for the storage of fresh meats, while simultaneously fulfilling the customers’ wishes for a more environmentally friendly packaging alternative.

## 1. Introduction

Fresh meat is considered a perishable food [1,2] that provides ideal conditions for microbial growth, with some organisms simply causing spoilage while others pose a serious threat to human health, making it a public health concern [3,4]. Meat safety can be challenged in various ways, e.g. chemical residues such as pesticides and antibiotics, diseases in animals such as transmissible spongiform encephalopathy and microbial contamination with pathogenic microbes or their toxins [5]. In any case, the contaminated meat must be removed from the food chain and consequently becomes waste, leading to an additional environmental burden. Meat production and consumption have been under additional scrutiny, not only for the public health issue, but also for their environmental impact and their role in microbiological safety [6]. Meat products generally are considered valuable foodstuffs, so the aim is to achieve the longest possible shelf life. This can be supported by economically and ecologically sensible packaging. The production of high-quality meat has a long tradition in the Tyrol/Bavaria border region. Nowadays, meat products are packaged and sold primarily in supermarkets instead of fresh over the counter at the butcher shop. In this study, the matrix pork was chosen due to its favored status by both producers and consumers within the European Union. According to the Organisation for Economic Co-operation and Development (OECD), pork is one of the most consumed meats over the entire globe, with the European Union and China favoring pork over any other meat, according to most recent data, over 30 kg of pork consumption per capita, respectively [7]. In 2014, pork was the most significant meat category in the European Union, with over 22.1 million metric tons produced, attributing to 51% of the annual production of all meats [8].

Food packaging as means of protection against product deterioration through damage to physical, chemical and biological properties has led to advancements in the choice of materials and packaging modifications in the last decades. Different packaging systems allow for different modulations of the gaseous environment of the packaged goods. Vacuum-skin packaging (VSP) relies on negative pressure to remove ambient air with sealing of pouches or rollstock formed packages to maintain the vacuum state [9]. Modified atmosphere packaging (MAP), on the other hand, changes the composition of the gaseous environment around the meat [4] through which the microflora of meat can be modulated, provided the cold chain is maintained throughout storage and transport, without any additives or other interventions. The aforementioned packaging systems and their varying gaseous environments are significant for packaging fresh meat.

The visual and packaging differences in VSP and BioMAP/MAP are illustrated in Figure 1. In the case of the MAP example, BioMAP would appear similar as only different packaging materials are used to achieve a similar packaging look and feel as MAP. As can be observed, with VSP, the packaged goods are sealed across their entire surface on a dimensionally stable bottom film or a prefabricated tray with a special skin film [10]. VSP relies on negative pressure to remove ambient air [9], eliminating the air around the meat [11]. In the case of MAP, the natural atmosphere in the package is removed [9] and replaced by a modified atmosphere (gas mixture) adapted to the packaged goods in order to maintain the shape, color and freshness of the product [12].

The impact of the different packaging atmospheres of VSP and BioMAP/MAP on the meat color is described in the following paragraphs.

When myoglobin is oxygenated, as is the case in high-oxygen MAP, it forms oxymyoglobin (OxyMb) that gives meat a stable red color until the reducing capacity is exhausted, and the pigments change to brown metmyoglobin (MetMb) pigments [9,14,15]. OxyMb is the reduced pigment form (Fe^2+^), in which O_2_ occupies the ligand position and the perceived color is red. MetMb is the oxidized pigment state of myoglobin, the dominant sarcoplasmic pigment in muscle, and the Fe^3+^ results in a brown or gray meat color [16]. In the absence of O_2_, myoglobin is reduced to deoxymyoglobin (DeoxyMb), changing the meat color to purple [16], as is the case in VSP. DeoxyMb is the reduced form of myoglobin (Fe^2+^), which creates a purple color in the absence of O_2_ [16]. The redox interconversions of myoglobin are described schematically in Figure 2.

The addition of CO creates a red pigment, carboxymyoglobin, which is highly stable for longer time periods than OxyMb. The use of CO in food industry, however, is controversial, with some countries (US, Canada, Australia and New Zealand) approving its application, while EU member states banned it from food processing. CO has been previously reported to mask meat spoilage, which was the primary concern raised for its prohibition as this may mislead consumers [19]. The addition of CO_2_ to the packaging atmosphere is used for microbial inhibition [20], while N_2_ is used as an inert filler gas to either reduce the proportions of other gases or to maintain pack shape [21]. In retail meat packaging, the modified atmosphere is usually comprised of 70–80% oxygen (O_2_) to provide a stable bloomed red meat color [22] and 20–30% carbon dioxide (CO_2_) to prolong the shelf-life by inhibiting bacterial growth [23].

It has been proven that the visual appearance and expected quality of meat are strongly related, meaning that the quality expected by consumers is largely inferred from intrinsic quality cues, such as meat color [24], which has been shown to be one of the most important fresh meat characteristics at the point of purchase [25,26,27]. While color changes in meat are not necessarily harmful and do not have to denote spoilage, they are considered undesirable by customers [23]. These consumer perceptions and evaluations of meat quality attributes can be affected by the type of fresh meat packaging system [28]. The color change in fresh meat is caused by a change in the protein myoglobin [29], which is the color pigment in a muscle and responsible for binding oxygen. The most well-known packaging materials, utilized by the food industry for over 50 years, are polyethylene (PE)- or copolymer-based materials [30]. The use of plastics has many advantages in comparison to other materials [31], such as being flowable and moldable to make sheets, shapes and structures while remaining lightweight; being generally inert, though not necessarily impermeable; being cost effective; and providing choices in respect of transparency, color, heat sealing and resistance and barrier [32]. However, their drawbacks are becoming more apparent with extended use of plastics. Since the majority of materials in the packaging industry are produced from fossil fuels, they are practically non-degradable [33], awareness of the environmental damage associated with conventional packaging has been growing [34]. The damage caused by the large amounts of packaging can be illustrated by the per capita production of packaging waste in developed countries, with roughly 169 kg of packaging waste per year and capita in the European Union in 2016 [35] and 220 kg of packaging waste per year and capita in the United States of America in 2015 [36].

Ultimately, in today’s society, packaging must meet both essential product requirements and specific environmental objectives [37]. Alternative to materials produced from fossil fuels, biodegradable materials, such as bioplastics, are being used for packaging various foods, with the aim of reducing the overall use of packaging materials derived from fossil fuels. Bioplastics play a key role in this transition by replacing the aforementioned fossil fuel-based packaging materials [38]. It has been found that consumers perceive bioplastics as more-sustainable packaging [39], closely relating it to other benefits such as naturalness and healthiness [40,41,42] and better taste [43] of the packaged product, higher cost [44] and overall increased quality [41]. It has been demonstrated that consumers are willing to pay more for high quality foods. Increased awareness of environmental issues has also generated a receptiveness to pay more for food packaged in biobased materials, which are considered friendlier to the environment [22].

Polylactic acid (PLA) is a biodegradable polymer, made primarily from renewable agricultural resources (e.g., corn) by fermentation of starch and condensation of lactic acid [45]. PLA is composed of lactic acid chains and exhibits tensile strength comparable to other commercially available polymers. In addition to its strength, biodegradability and compostability, it also demonstrates a high resistance to fossil fuel-based products, is sealable at lower temperatures and acts as both flavor and odor barrier for foodstuffs [45,46]. Although synthetic polymers are gradually being replaced by biodegradable materials, especially those derived from replenishable, natural resources [47], and despite the promising characteristics of bioplastics such as PLA, our analysis of the available literature indicates only a limited number of studies is available focusing on the use of biopolymer for food packaging applications, especially for fresh meat. This is despite the fact that their use would represent a real advancement in reducing environmental pollution [48].

The aim of this study was to evaluate the sustainability of BioMAP as compared to MAP by investigating the life span of pork products packaged in either of the aforementioned packaging methods. This should be accomplished by assessing the sensory aspects and thus mimicking consumer acceptance, evaluating changes in meat color, as well as the microbiological attributes of the packaged fresh pork products. The results show no significant differences (*p* > 0.05) in ground pork and pork loin stored in BioMAP and conventional MAP for the evaluated sensory attributes, meat color or TBC over extended storage times

## 2. Materials and Methods

### 2.1. Raw Materials and Storage Conditions

In this study, ground pork (GM) and boneless pork loin (*longissimus dorsi*) (PL) with subcutaneous fat and rind, from both male and female pigs of the two highest meat classifications in terms of lean meat and muscle development [49], were used, with an overall average lean meat content of above 55%.

The animals were slaughtered according to standard practice and regulations at a slaughterhouse, with the GM samples subsequently being ground on site, using a hole-plate 5 mm in diameter. Following slaughter and/or grounding, the meat samples were packaged in vacuum bags and transported at less than 4 °C to the packaging site of MULTIVAC Sepp Haggenmüller SE & Co. KG in Wolfertschwenden, Germany. Upon arrival, the vacuum bags were opened and repackaged in equally sized portions while keeping direct human contact with the meat to a minimum in order to avoid microbiological contamination. Both the PLs and the GM were equally split between the two packaging types and packaged accordingly. Once packaged, the samples were transported at less than 4 °C to Management Center Innsbruck for final storage. Upon arrival in Innsbruck, samples were immediately transferred to a humidity-controlled environment at 4 ± 1 °C and stored unstacked until used for the experiment. On the days of analysis, samples were removed accordingly while ensuring the overall temperature and humidity environment were not affected.

### 2.2. Packaging Materials

The trays for MAP were made from mono-amorphous polyethylene terephthalate (PET; MAPET^®^II) (Færch Plast Group, Holstebro, DK), with sealing top films made from a PET, laminate adhesive, PE/ethylene-vinyl alcohol-copolymer (EVEP) layer combination (Südpack, Ochsenhausen, Germany). For BioMAP, trays were made from PLA resin (NatureWorks, Minnetonka, MN, United States), with sealing top films made from a layer combination of cellulose and PLA (NATURABIOMAT GmbH, Schwaz, Austria). Oxygen-transmission rates (OTR) and water-vapor transmission rates (WVTR) for both trays and top films were obtained by the manufacturer for MAP and tested prior to any performed experiments for BioMAP. Heat resistances for both packaging types were obtained through manufacturer-provided documentation. The potential impact of temperature and humidity on the material durability of PLA are addressed in Table 1, which shows the respective OTRs, WVTRs and heat resistances for both MAP and BioMAP.

The heat resistances of the used materials are comparable and thus material durability is not expected to be diminished in either of the packaging types resulting from the temperatures used. Similarly, no negative impact is expected on the humidity and temperatures used during transport and storage of the samples. This is of importance especially for the BioMAP so that conditions used in active composting are avoided that could risk a potential degradation of the packaging materials.

A gas composition of 70% O_2_ and 30% CO_2_ was chosen for both packaging types.

### 2.3. Sensory Analyses

Sensory analysis of GM and PLs was always performed by those with an identical scientific education but being sensory untrained to simulate how consumers would perceive the tested meats in a retail setting.

The sensory attributes were evaluated for both packaged and unpackaged meat samples following the check-all-that-apply (CATA) scheme with additional intensity scores from 0 (no intensity) to 5 (maximum intensity) attributed to each assessed characteristic. This can be compared to the rate-all-that-apply (RATA) scheme as described by others [50,51,52]. As previously mentioned, the assessment type was chosen to simulate consumer perception of packaged meats in a retail setting, as well as perception of unpackaged meat in a home setting following the meat purchase.

For the packaged sensory evaluation, the packages were taken from storage on the days of analysis and immediately evaluated for their sensory attributes. The attributes evaluated in GM were liquid discharge (no/maximum discharge), old or spoiled appearance of meat (no maximum spoiled appearance), product presentation (unpleasant/very pleasant presentation), overall impression (negative/very positive impression) and color attributes of the meat (gray, green, dark, light, red, violet and brown; no coloration/strong coloration of the aforementioned colors). The evaluated attributes in PL were liquid, old or spoiled appearance of meat, product presentation and color attributes of the meat (dark, light, red, violet and brown)—the RATA scores were analogous to the ones used in packaged samples. For both GM and PL, there was also an opportunity to note comments about the meat samples if they were needed in addition to the standardized assessment; these data were not used in any analysis.

For the unpackaged sensory evaluation, the opened packages were assessed within a maximum of 15 min of opening. The attributes evaluated were analogous to the packaged attributes in GM, as described above. For PL, the following attributes were assessed: color attributes of the meat (dark, light, red, violet and brown; no coloration/strong coloration of the aforementioned colors), overall impression (negative/very positive impression) and odor attributes (sour, rancid, musty and sweet; no/very strong odor). Analogous to the packaged evaluation, there was an opportunity to note comments if needed; these data were not used in any analysis.

To test the accuracy of the sensory analysis and determine differences between packaging systems, the same meat matrices were packaged in VSP and tested analogously to the MAP and BioMAP (packaged and unpackaged). The same approach as described above for MAP and BioMAP was used for VSP and the results were compared to both MAP and BioMAP to determine potential differences.

### 2.4. Total Bacterial Count (TBC)

Microbiological analyses were performed to evaluate the hygiene process indicator microorganisms. TBC was assessed by using the horizontal pour plate method for enumeration, according to ISO 4833-1:2013 [53]. For each sample, 10 g of meat and 90 mL buffered peptone water (Carl Roth, Karlsruhe, Germany) were homogenized in a stomacher (BagMixer^®^ 400 P, Interscience, Saint Nom la Breteche, France) for 2 min at room temperature. Appropriate serial dilutions of the homogenate were then made using buffered peptone water, with 1 mL of each dilution being plated on PCA (Plate Count Agar, Carl Roth, Karlsruhe, Germany). The plates were incubated at 30 °C for 72 h to determine TBC. Results were expressed as log_10_ cfu/g and performed in triplicate.

For both meat matrices, the microbiological limits “m” and “M”, as laid out by the European Union for TBC, were considered as indicators of microbiological load [54].

### 2.5. Color Analyses

The meat color was analyzed using the CIE (Commission Internationale de L’Eclairage) L*a*b*-coordinate system (lightness (L*), redness (a*), yellowness (b*)) [55] with a colorimeter CM-5 (Konica Minolta Inc., Osaka, Japan) at an observer angle of 10°, with a measurement area diameter of 30 mm. The colorimeter was calibrated on each measurement date according to the calibration of the instrument’s manufacturer with standardized white and black calibration plates. The color coordinates were obtained within a maximum of 150 min between opening of the meat packages and taking the measurements for both GM and PL. Color measurements were made using 692.104-BF cylindrical cells 40.5 mm × 60 mm (Hellma GmbH & Co. KG, Müllheim, Germany). The measurements were taken in a controlled room without daylight or direct exposure to artificial light.

For analyses of GM, all available meat was first mixed by hand to ensure equal distribution of fat and meat. Approximately 30 g of meat were weighed into the cuvettes and distributed evenly to ensure equal distribution of meat throughout. For analyses of the PLs, cylinders with a 2.5 cm diameter were cut from the meat by using a cork drill. The samples were taken from the mid-section of the meat, keeping a 2 cm distance between each sample. For the measurement, one cylinder was placed into the middle of the cuvette to ensure full coverage of the measurement area by the meat. For each sample, three measurements were performed. Each measurement was repeated after turning the cuvette by 90° between the two measurements, resulting in a total number of six measurements per sample. The average of the six readings was used for further statistical analyses.

### 2.6. Color Distance (∆E)

The color distance, also known as Delta E (∆E), between two colors (L*, a*, b*)_v_ and (L*, a*, b*)_p_ is calculated as the Euclidean distance according to ISO/CIE 11664-4:2019 [56]. It represents the perceived magnitude of color differences [56]. The formula used to calculate ∆E is shown in the Equation (1), below:(1)$$∆E_{p,v}=\sqrt{{\left(L_{p}^{*}-L_{v}^{*}\right)}^{2}+{\left(a_{p}^{*}-a_{v}^{*}\right)}^{2}+{\left(b_{p}^{*}-b_{v}^{*}\right)}^{2}}$$

To determine any perceived color differences for the (un-)experienced observer, the color distance between packaging types on the measurement days, as well as color differences within packages between measurement days, was calculated. This was done to simulate consumer perception beyond the sensory analysis described in Section 2.3. The calculated ∆E values were then classified in Table 2 by the following ranges [57]:

## 3. Results

### 3.1. Sensory Analyses

The average sensory attributes were assessed as described in Section 2.3. The attributes are shown on the x-axes of Figure 3 and Figure 4 in shortened form and are described in the Appendix A
Table A1.

The statistical analysis in GM showed no significant differences (*p* > 0.05) between the meat packaged in BioMAP and MAP, as can be seen in Figure 3A,B.

In both packaging materials, the liquid discharge increased after 10–13 days of meat age (packaged or opened), with gray coloration of the meat increasing in both materials at similar meat ages. In both packaging types, GM did not develop any violet coloration (packaged or opened). The average sensory evaluations of packaged and opened GM in VSP can be seen in Figure 3C. Sensory evaluations in BioMAP/MAP showed a significant difference (*p* < 0.05) for the attributes liquid discharge (packaged or opened), meat lightness (packaged or opened), violet and brown coloration of meat (packaged) and gray coloration of meat (opened).

The graphical representation of the average sensory attributes evaluated in packaged and opened PL in BioMAP and MAP can be seen in Figure 4A,B and for VSP in Figure 4C. The values shown were selected to facilitate easier comparison with GM-data. Full data can be found in the Appendix A
Figure A1.

The sensory analyses in PL revealed significant differences (*p* < 0.05) in the dark coloration of meat between samples packaged in BioMAP and MAP. While a dark coloration of meat was observed in samples packaged in BioMAP and MAP, the samples in BioMAP showed an overall darker coloration starting at day ten of meat age and continued this trend throughout the assessed time. No other significant differences (*p* > 0.05) between the samples in BioMAP and MAP were noticed, neither with packaged nor opened samples. In both packaging types, PL did not develop any significant violet coloration (packaged or opened), and brown coloration appeared to increase slightly after 21 days (packaged or opened). The off-odors increased in both packaging types after 14 days of meat age, with sweet and sour smells appearing to be the strongest odors assessed. When comparing PL in BioMAP and VSP, significant differences (*p* < 0.05) were found for all attributes except dark coloration and overall impression in packaged and opened samples, light coloration in packaged samples and red and brown coloration in opened samples. The comparison of PL in MAP and VSP showed significant differences (*p* < 0.05) for the violet coloration of the meat in packaged and opened samples, as well as brown coloration in packaged samples. Additionally, significant differences (*p* < 0.05) were seen in the lightness of opened samples, as well as musty and sweet odors.

No difference in the sensory properties of pork meat packaged in MAP and BioMAP could be noted, except for the meat darkness in PL.

### 3.2. Total Bacterial Count (TBC)

The analyses of TBC in GM and PL packaged in BioMAP and MAP did not reveal any significant differences (*p* > 0.05) over the analyzed 34 days of meat age in GM and 52 days of meat age in PL. The slopes of the fitted curves did not differ significantly (*p* > 0.05) between GM and PL in BioMAP and MAP.

The initial TBC of GM (Figure 5A) was 1.99 log_10_ cfu/g in BioMAP and 2.25 log_10_ cfu/g in MAP and increased to 12.20 and 12.22 log_10_ cfu/g, respectively, until the measurement endpoint of 34 days of meat age. The lower microbial limit “m” [54] was reached in both BioMAP and MAP on Day 9 of meat age, whereas the higher microbial limit “M” [54] was reached for both tested packaging types at Day 13 of meat age.

The graphical representation of the evaluated TBC in PL in BioMAP and MAP is available in Figure 5B. The depicted values were selected to facilitate easier comparison with GM-data. Full data can be found in the Appendix A
Figure A2.

The initial TBC in PL (Figure 5B) was 0.65 log_10_ cfu/g in BioMAP and 0.00 log_10_ cfu/g in MAP and increased to 6.89 l and 6.47 log_10_ cfu/g, respectively, until the experimental endpoint of 52 days of meat age. The initial microbial load influenced the further development of TBC over the evaluated time; PL in BioMAP reached “m” on Day 13 and “M” on Day 16 of meat age, whereas PL in MAP reached the microbial limits three days later on Days 16 and 19, respectively.

### 3.3. Color Analyses

The average L*-, a*- and b*-values in GM and PL stored in BioMAP and MAP can be seen in Figure 6.

There was no significant difference (*p* > 0.05) in L*-values for either GM or PL stored in BioMAP and MAP, as can be seen in Figure 6A. The initial L*-values of GM in BioMAP increased from 57.15 to 58.39 between Days 4 and 24 of meat age, whereas L*-values of GM in MAP increased from 57.33 to 60.27 during the same period. In both packaging types, L*-values appeared to be increasing and decreasing between measurement points in GM, which may be a result of the meat matrices used in the various experimental runs. However, these observed changes in meat lightness were similar between the packaging types. In PL, storage in BioMAP led to L*-values between 52.50 and 52.30 from Day 4 to 52 in meat age, whereas L*-values in MAP were measured at 49.39 on Day 4 and 55.08 on Day 52 of meat age (full data in the Appendix A in Figure A3). In BioMAP, the initial L*-value decreased after Day 7 in meat age and proceeded to increase again until Day 28. The lightness then decreased again and remained relatively stable between Days 31 and 52 of meat age (see Figure A3 for full data). In MAP, the increase in lightness took place gradually over the measurement duration with no major drops in L*-values as seen in BioMAP. The increase in L*-values throughout the storage duration has been noted by others previously [58,59]. The increase in lightness may be explained by the highly oxidizing conditions in BioMAP/MAP, leading to changes in meat structure such as protein conformational changes [60] and thus greater light dispersion.

There was no significant difference (*p* > 0.05) between a*- and b*-values in either GM or PL stored in BioMAP and MAP, as shown in Figure 6B, C. In GM, the average measured a*-value in BioMAP decreased from 15.75 on Day 6 to 12.10 on Day 34 of meat age. In MAP, the average a*-value decreased from 16.89 on Day 4 to 10.36 on Day 34 of meat age. In PL, a*-values in BioMAP decreased from 10.00 to 5.56 between Days 4 and 52 of meat age, in MAP the values decreased from 9.12 to 5.32 in the same time frame (see Figure A3 for full data). While a*-values decreased in both packaging types over the storage duration, the changes were not significantly different (*p* > 0.05) between packaging materials. This confirms the results of Panseri et al. [1], who did not find any significant differences between red meat stored in PLA-based packaging and polyethylene terephthalate/polyethylene (PET/PE) based packaging and found PLA-based packaging to be as suitable to maintain the red color in red meat as the PET/PE-based packaging. The initial cherry-red visual color was expected due to the formation of oxymyoglobin in the high oxygen packaging atmosphere [61], whereas a subsequent decrease of redness has been associated with the formation of metmyoglobin [62], which leads to an undesirable brown color (Figure 2). The decrease in surface a*-values throughout storage has also been reported by others in various meat matrices [23,58,59,63]. Our findings confirm the results obtained by Jayasingh et al. [64], who reported that beef in high oxygen MAP maintained its bright red color for ten days. However, the initial intense red color only lasted a few days, as was also shown by Martínez et al. [65], in fresh pork sausages.

In GM, the average b*-values in BioMAP changed from 14.53 to 12.69 between Days 4 and 34. In MAP, the b*-values changed from 14.15 to 14.35 in the same period. In PL, the b*-values change from 10.47 to 9.19 in BioMAP between Days 4 and 52 of meat age. In MAP, the measured values on Day 4 and 52 were both 8.88 (see Figure A3 for full data).

### 3.4. Color Distance (∆E)

The color distance (∆E) was chosen as an additional parameter as it can show color differences by accounting for combined changes in the L*a*b* color space, and thus might be a useful tool [18] to indicate consumer acceptance (Figure 7).

The color distance (∆E) is—as described in Equation (1)—the Euclidean distance between two colors, which are comprised of their respective L*-, a*- and b*-values. The higher is the ∆E result, the greater is the relative color change compared to the original meat color [61]. The ∆E values were calculated based on Equation (1) and evaluated according to the scaling shown in Table 2. Any major fluctuations in the ∆E values can be explained as they are directly related to the individual L*-, a*- and b*-values of the specific measurement days.

The average calculated color distance values for both GM and PL in BioMAP and MAP are indicated in the Appendix A in Figure A4. In this study, we did not find significant differences (*p* > 0.05) between ∆E values in either GM or PL in BioMAP and MAP. Figure 7 indicates the differences between ∆E scores of GM and PL in MAP and BioMAP. The color ranges indicated are based on Table 2. It is noticeable that solely on Day 34 of meat age a difference between GM in MAP and BioMAP can be discerned by an unexperienced observer, whereas differences on Day 14 of meat age are only noticeable by an experienced observer. In PL, the differences on Days 7 and 28 of meat age are barely noticeable by an experienced observer.

## 4. Discussion

### 4.1. Sensory Analyses

As mentioned in Section 3.1., there was no difference in the sensory properties of pork meat packaged in MAP and BioMAP, except for the meat darkness in PL. However, compared to VSP, it appears that the packaging system had an impact on these properties. Especially oxygen-dependent parameters, such as color, were impacted, as has also been evidenced by others [11,58]. When comparing the sensory attributes of GM and PL packaged in BioMAP/MAP and VSP (Figure 3 and Figure 4), the impact of the used packaging systems on meat coloration becomes apparent. The differences in meat coloration in packaged GM and PL can be explained by the available O_2_ in the packaging. While BioMAP/MAP had a modified atmosphere containing 70% O_2_, VSP contained close to no O_2_, resulting in the differences in perceived meat color, owing to the oxygenation of the myoglobin (Figure 2). Additionally, it can be considered that liquid discharge could be directly related to the packaging system for GM. This sensory attribute can be linked to the drip loss of meat, which has been shown to increase for meat stored in MAP when compared to VSP [15,66]. This could be explained by degradation of desmin through proteolysis [67], or protein crosslinking, which might also affect the drip loss in meat [68].

Off-odors in PL in BioMAP/MAP, compared to PL in VSP, can be explained by the availability of packaging systems and the O_2_ availability in BioMAP/MAP. This development of off-odors over time has been shown by others in the various meat matrices [23,63,65]. Martínez et al. theorized that off-odor-causing products of lipid oxidation are retained within the package in MAP, thus allowing consumers to perceive off-odors when packages containing oxidized meat are opened [65]. The onset of rancid odors in BioMAP/MAP could be explained by increased lipid oxidation due to the presence of O_2_ in the packaging, when compared to samples in VSP. The authors of [69] showed that musty odors in meat can be caused by the presence of certain ketones, which have been proven to be created by the presence of *Pseudomonas* spp. [70]. The aerobic conditions in BioMAP/MAP are beneficial for the growth of *Pseudomonas* spp. [71] and explain the presence of musty odors in PL in BioMAP/MAP, not however in VSP over the assessed timespan. Similarly, it was shown that certain ketones and aldehydes are responsible for sweet odors in the meat [69]. The presence of ketones in aerobic conditions can be explained by the presence of *Pseudomonas* spp., whereas certain aldehydes are created, for example, by *Carnobacterium* spp. and *Brochothrix thermosphacta* [70]. The growth of *Carnobacterium* spp. has been found to be significantly influenced by storage conditions, namely the O_2_ availability during storage [72]. *Brochothrix thermosphacta* has been found to have the ability to produce acetoin as a major end product in spoilage under aerobic conditions, using glucose as substrate. This compound has been associated with sweet odor in meat [73].

### 4.2. Total Bacterial Count (TBC)

The trend of initial microbial load being inversely proportional to shelf life (Figure 5) has been noted by others [23,74] in various meat matrices and confirms our finding that microbial load is one of the most important parameters determining meat shelf life [75] in pork [76].

Our findings are in agreement with Panseri et al., who did not find any significant differences in the microbiological parameters investigated for red meat stored up to eight days in MAP made from PLA or PET/PE [1]. Little research exists otherwise investigating potential differences between conventional MAP and MAP using biodegradable materials for meat packaging.

### 4.3. Color Analyses

In both packaging types, the b*-values appeared to be increasing and decreasing between measurement points in GM, which might be a result of the meat matrices used in the various experimental runs. However, these observed changes in b*-values were similar between the packaging types. Our findings do not confirm findings in ostrich meat, where a continuous decrease in MAP was seen over the storage time [61,63], but are in line with the findings of others in various meat matrices, such as foal meat [23] or beef steaks [15], the latter study observing a relatively constant or slightly decreased b*-value in beef steaks following an initial increase.

However, as noted by Mancini et al., it must be considered that colors represented by the b*-value measurements are not typically or intuitively related to meat, whereas L*- and a*-values are straightforward and can easily be applied to muscle color [18]. The significance of b*-values to describe any differences in meat characteristics can thus be questioned and one might refrain from weighting.

The differences in the initial L*-, a*- and b*-values between GM and PL can most likely be explained by the increased rates of oxygenation and the resulting increase of MetMb in GM samples. It has been shown that with increasing myoglobin oxidation, pork becomes darker and redder [77]. Oxygenation of GM due to the mincing process influenced L* significantly [77], as myoglobin oxidation is more prominent in GM. This is due to its nature of having an increased exposed surface as compared to PL.

### 4.4. Color Distance (∆E)

As no significant differences (*p* > 0.05) could be identified between the other color measurements (L*, a*, b*), these results were to be expected. In terms of consumer acceptance, the ∆E values—and the differences between them—put into perspective whether any differences in color were to be observed between packaging types on the various measurement, and, if so, how large the perceived differences would be. Meat color is one of the first quality attributes perceived by the consumer, who relies on it as an indication of freshness and wholesomeness. The color preference is sufficient enough to influence the likelihood of purchasing meat [78]. However, it has been demonstrated that color preferences vary internationally, with consumers worldwide being equally distributed for having preferences for either darker or lighter meat [26]. To predict the acceptability of pork packaged in BioMAP as compared to MAP by consumers worldwide, it was necessary to compare potential differences in color distances between the packaged meats. Therefore, it was determined whether differences between the meats in both packaging types were visible to the consumer and influenced their purchasing preference for either of them, rather than merely considering any differences in meat coloration by itself.

Generally, as an important aspect of sensory analysis it should be pointed out that meat quality is the sum of all sensory, nutritional, hygienic and processing properties of meat [79]. Additionally, challenges with processing sensory tests for meat can arise due to animal-to-animal variation, variation within the same animal and variation in the used animal sections. Moreover, the way the meat is prepared prior to the test is also crucial [80]. In this study, we opted to approach these challenges by a conscious choice of analyzed meat quality and used tests. The sensory aspects of meat quality were addressed through evaluation of sensory attributes in packaged and opened meat samples for both meat matrices. By choosing samples from the top two meat classes and with similar fat content, we attempted to minimize the influence of nutritional aspects on the performed tests. The hygienic and processing aspects were addressed by analyzing microbiological load, which did not show significant differences (*p* > 0.05) over the tested timespan either. In general, it can be said that the sensory data collected can be verified by the instrumental color analyses.

## 5. Conclusions

This study aimed to evaluate the sustainability of BioMAP compared to MAP by investigating the life span of pork products packaged in either of the aforementioned packaging materials. We investigated the use of PLA- and cellulose-based packaging as a sustainable alternative to conventional fossil fuel-based packaging, with the premise to preserve fresh pork products as well as, or better than, conventional packaging. This should be accomplished by assessing sensory aspects, mimicking consumer acceptance, evaluating physicochemical changes such as meat color and analyzing microbiological attributes of the packaged fresh pork products over an extended storage period greatly exceeding the minimum shelf life that is commonly assigned to these products.

The biodegradable packaging did not show any substantial disadvantages compared to conventional packaging for the storage of ground pork and pork loin over 34 and 52 days of meat age, respectively. It can be assumed that the microbiological shelf life was not influenced by the packaging materials used, but rather the meat quality of the samples, as well as hygiene standards kept prior to and during packaging of the raw meats. The results demonstrate that meat quality is more influenced by the presence of oxygen, as was shown through direct comparison of meats stored in BioMAP/MAP and VSP, than the packaging materials used in BioMAP and MAP. While significant differences (*p* < 0.05) between the packaging systems BioMAP/MAP and VSP were seen in the sensory evaluation of packaged and opened pork meat products, no significant differences (*p* > 0.05) were found within the MAP systems using different packaging materials.

As with all new or novel packaging developments destined for consumers, cost, organoleptic, consumer preference, toxicological, safety and regulatory considerations must be addressed for the use of PLA if this type of technology is to be adopted and expanded on a larger scale [81].

PLA is suitable for food contact applications and approved for use in food packaging, including direct contact applications, with its classification being generally recognized as safe (GRAS) [82]. It complies with a number of global food contact regulations [83] for food applications, according to the Directive 1935/2004 [84]. Currently, bioplastics represent only roughly 1% of the annual global plastic production, although with rising demand and the development of more sophisticated biopolymers a continuous market growth is projected. PLA has, besides starch blends, the highest production capacities out of the biodegradable materials available [85]. The number of studies evaluating the use of biodegradable packaging for storage of fresh meat is, to the best of our knowledge, limited. Further studies are necessary to establish biodegradable materials as a viable alternative to commonly used, fossil fuel-based materials for other food products as well as other commodities. The aspects of recyclability need to be further addressed if a large-scale roll-out of PLA-based packaging is planned. Additionally, the issue of price must be addressed as existing products are currently exclusive [22]. It has been found that consumers are willing to pay a premium price for food products displaying signals of sustainability due to a presumed higher quality of the product [41]. However, in this study, we were not able to display a superior quality of meat products packaged in BioMAP compared to MAP, but rather the quality was equitable. The question thus remains whether consumers would still be willing to pay a premium price for sustainable packaging if no added benefit to product quality was given. For producers of green packaging the issue of risking ongoing expenses for green packaging, reasonable containment of costs for green packaging and recuperation of said costs in a reasonable time period remain [86].

With this study, we were able to show that PLA packaging is suitable for the packaging of fresh pork products. It is also possible that, with adequate changes to the current systems in place and an adjustment of price towards that of non-renewable material, PLA can be an equivalent alternative to conventional packaging made from fossil fuel-based materials. Further study will calculate their environmental impact based, e.g., on life-cycle assessments. Previous meta-analysis has shown an increased development over the last years and predicted further studies in this field [87].

## Figures and Tables

**Figure 1 foods-09-01802-f001:**
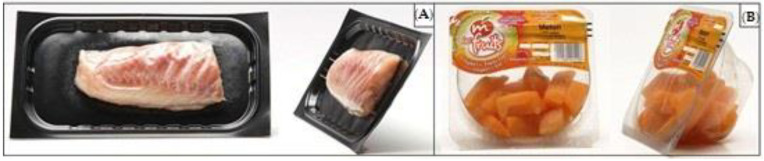
Illustrations of (**A**) Vacuum-skin packaging (VSP) and (**B**) biodegradable modified atmosphere packaging (BioMAP) and modified atmosphere packaging (MAP) systems using fish and fruit as examples of packaged goods. Source: [13].

**Figure 2 foods-09-01802-f002:**
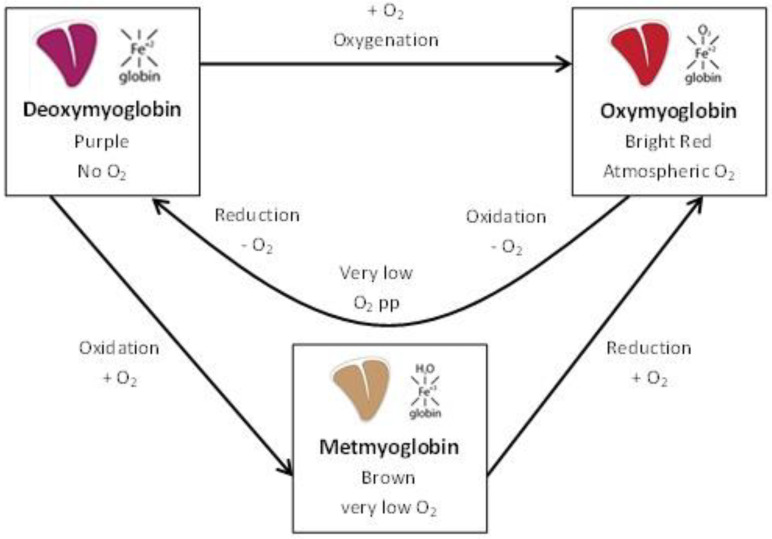
Visible myoglobin redox interconversions on the surface of meat. Inspired by [17,18].

**Figure 3 foods-09-01802-f003:**
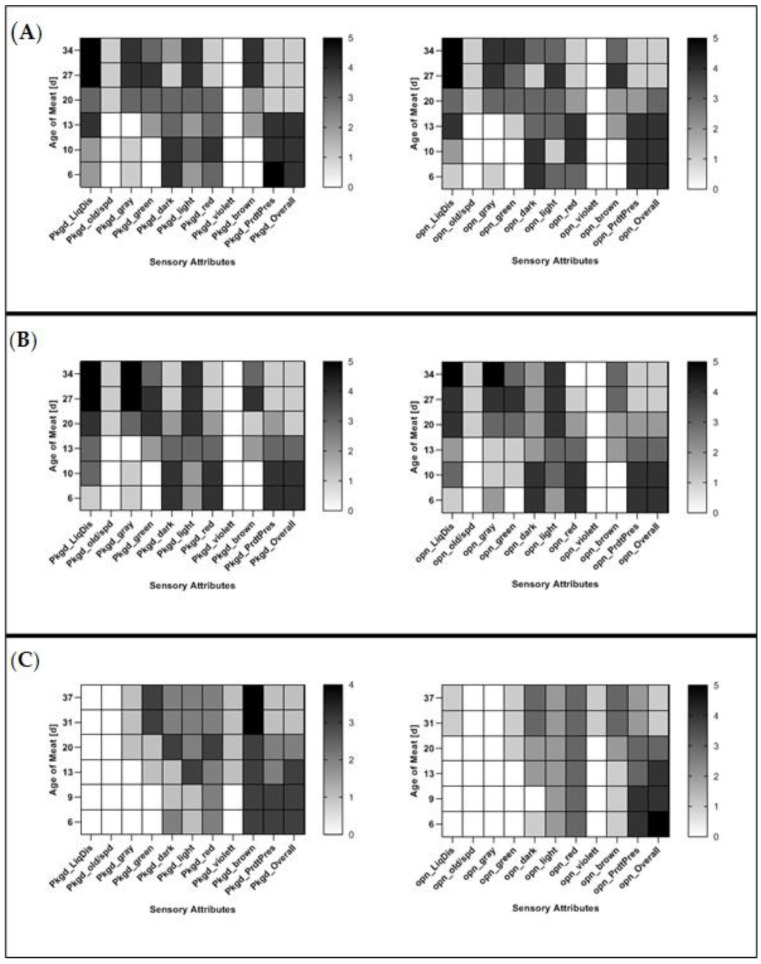
Average sensory evaluation of packaged and opened ground pork (GM) in: (**A**) BioMAP and (**B**) MAP over a meat age of 34 days; and (**C**) VSP over a meat age of 37 days.

**Figure 4 foods-09-01802-f004:**
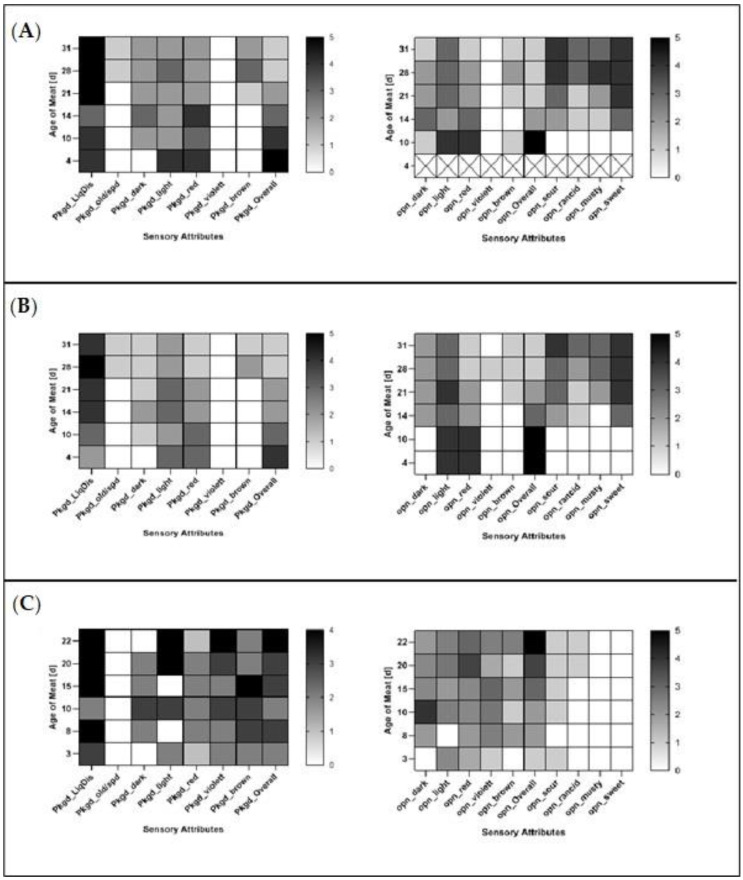
Average sensory evaluation of packaged and opened pork loin (PL) in: (**A**) BioMAP and (**B**) MAP over a meat age of 31 days; and (**C**) VSP over a meat age of 22 days.

**Figure 5 foods-09-01802-f005:**
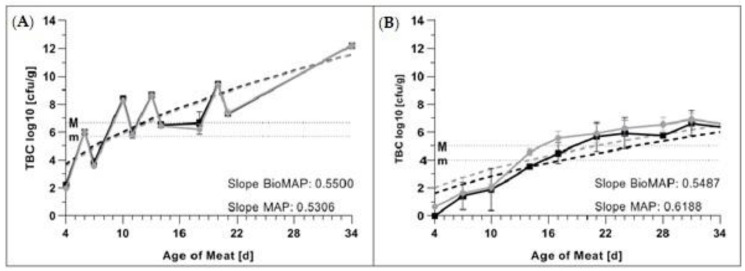
Average TBCs (±S.D.) in 

 BioMAP and 

 MAP in (**A**) ground pork (GM) and (**B**) pork loin (PL) over a meat age of 34 days. Microbiological limits “m” and “M” as per European Union legislation are indicated for both meat matrices. The fitted curves are indicated by dotted lines for GM and PL in BioMAP and MAP, respectively, with slope values included in the graphs.

**Figure 6 foods-09-01802-f006:**
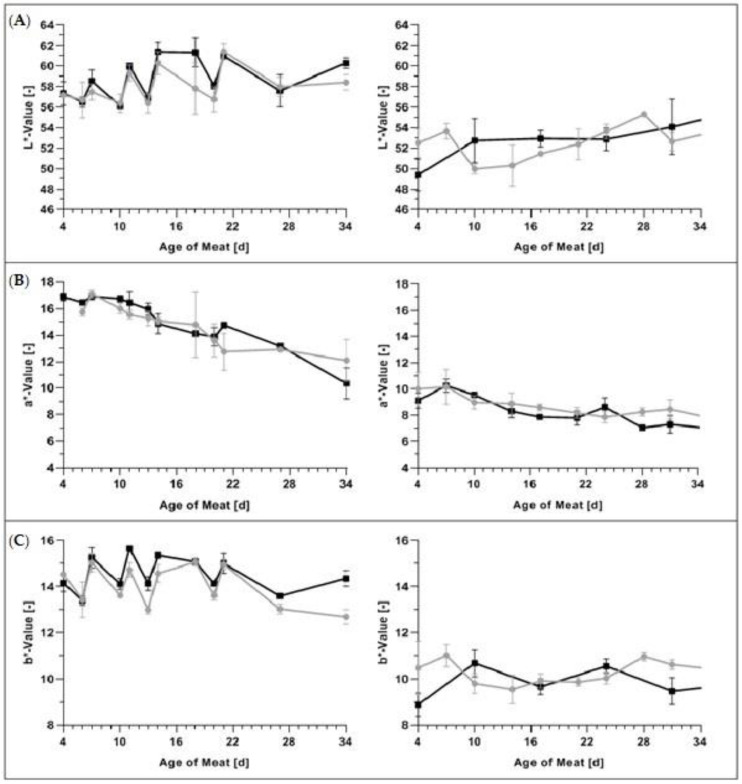
Average (**A**) L*-values (±S.D.); (**B**) a*-values (±S.D.); and (**C**) b*-values (±S.D.) in 

 BioMAP and 

 MAP in left-hand GM and right-hand PL over a meat age of 34 days.

**Figure 7 foods-09-01802-f007:**
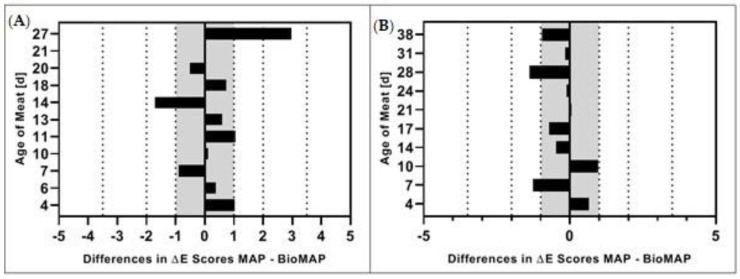
Differences in ∆E in MAP and BioMAP in (**A**) ground pork (GM) and (**B**) pork loin (PL) over meat ages of 27 and 38 days, respectively. The dotted lines indicate color ranges as described in Table 2, while the the gray areas indicate the range where no difference is noticeable.

**Table 1 foods-09-01802-t001:** OTRs, WVTRs and heat resistances for trays and top films of MAP and BioMAP.

Packaging Type	Trays			Top Films		
	OTR	WVTR	Heat Resistance	OTR	WVTR	Heat Resistance
MAP	100–160 ^1^ [cm^2^ × 25 µm/m^2^ × day × bar]	10–30 ^2^ [g × 25 µm/ day × m^2^]	−40−40 [°C]	<2.5 ^1^ [cm^3^/m^2^ × day × bar]	<6.0 ^3^ [g/24 h × m^2^]	10–30 [°C]
BioMAP	32.77 ^1^ [cm^3^/m^2^ × day × bar]	37.60 ^4^ [g/day × m^2^]	54–55 [°C]	1.44 ^1^ [cm^3^/m^2^ × day × bar]	11.33 ^4^ [g/day × m^2^]	15–30 [°C]

^1^ Measurements performed according to DIN 55380. ^2^ Measurements performed according to ASTM E-96. ^3^ Measurements performed according to ASTM F 1249. ^4^ Measurements performed according to DIN 53122-1. OTR, oxygen-transmission rates. WVTR, water-vapor transmission rates.

**Table 2 foods-09-01802-t002:** Color ranges for ∆E values.

0 < ∆E_p,v_ < 1	The difference is unnoticeable
1 < ∆E_p,v_ < 2	The difference is only noticeable by an experienced observer
2 < ∆E_p,v_ < 3.5	The difference is also noticeable by an unexperienced observer
3.5 < ∆E_p,v_ < 5	The difference is clearly noticeable
5 < ∆E_p,v_	Gives the impression that these are two different colors

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
