# Peer review of "Suitability of Biodegradable Materials in Comparison with Conventional Packaging Materials for the Storage of Fresh Pork Products over Extended Shelf-Life Periods"

_foods, 2020, doi:10.3390/foods9121802_

Round 1
Reviewer 1 Report
|
Please add in the introduction some references about the importance of the right choice of the packaging systems and their environmental evaluation such as: https://doi.org/10.2202/1556-3758.1687
https://doi.org/10.1007/978-981-287-913-4_1
|
|
Line 183, Paragraph “packaging materials”: I think you should be more specific about the material thickness, because also OTR and WVTR depend on the thickness of the packaging. In table 1 the units of measurement are not uniform for trays; for MAP the Oxygen Transmission Rate is calculated for an area (cm2) and a thickness of 25 micrometres, moreover which is the thickness for the BioMAP solution? The WVTR for BioMAP is calculated as weight capacity/m2, instead for MAP it is also considered the thickness of 25 micrometres, why? In top films they are uniform.
|
|
|
|
Line 283, line 299, line 333, line 398: “Error! Reference source not found”. It is a mistake.
|
|
Line 298: The figure 3c refers to GM, not PL. |
|
Which is the packaging material used for VSP?
|
|
Line 326-341: What do the initial conditions depend on? They depend, for example, on the quality of meat or the hygienic conditions and so on. You should also specify why “m” and “M” depend in the initial conditions for PL.
|
|
Line 328: Why do you choose 34 days as the measurement endpoint of meat age? I suggest you specify these choices for sensory analyses, TBC, color analyses and distance.
|
|
Line 408: Why is it noticeable that on day 34 the difference between GM in MAP and BioMAP? In figure 7 that difference is referred to the day 27. |
|
Line 424-426: the liquid discharge could be related to the packaging system for GM in MAP, but what about PL? I suggest you explain more in detail the relationship between the liquid discharge and the packaging system both for GM and LM with BioMAP/MAP and VSP.
|
|
Line 462-465: the colors represented by b*-value measurements are not related to meat; I suggest you deepen this issue.
|
Author Response
Comments to the Reviewer:
Thank you very much for reviewing our manuscript and all your helpful and delighting remarks. It´s a pleasure to bring all the perspective in one story.
All the question regarding the packing material used deals with different level of industrial confidentialness. The units of the OTRs, WVTRs and the OTRs, WVTRs, and heat resistances for trays and top films of MAP and BioMAP are summarized based on all original guidelines and we didn´t standardized the value related to these American and Europeans standards used.
We have to apologize to all the imprecise notifications. The manuscripts were check again linguistically, inaccuracies and on the content level. And interesting point of course will be the influence of the PLA based packaging in suitable the sustainable goals of the UN. LCA analysis could be a proper method to calculate their influence.
We upload the comparison of the revised and the edited version.

Reviewer 2 Report
Comments to the Authors
The subject of the article is very interesting. The paper describes a study to evaluate the shelf-life of fresh pork products packed with different packaging namely conventional and biodegradable materials. The authors evaluate different parameters including sensory analysis, total bacterial count and colour analyses.
The article is of great importance and interest in the field of food science and of great interest for food industry.
The objective of the paper is clearly defined.
However, it would be interesting if the authors include a Table summarizing the shelf-life of meat products packed with different materials.
It should also be interesting if the authors comment on other biodegradable materials besides PLA to be applied in the packaging of meat products.
Author Response
Thank you very much for reviewing our manuscript and all your remarks. Our focus in the submitted draft manuscript is related on the influence of PLA packaging materials on fresh pork products. And the idea for QualiMeat as third party funded project based on serval feasibility studies which analyse the impact fruit products (f.e. published in Mistriotis et al. 2016; https://doi.org/10.1016/j.postharvbio.2015.09.022).
Due to the various influences on the estimated shelf life of meat regarding to the TBC at the beginning it would be not possible to define a general guideline. That´s the reason that we include in all the relevant figures the microbial limits m/M defined by the EU legislation.
We attached the comparison of both.
